# When Helping Hurts: A Zemiological Analysis of a Child Protection Intervention in Adolescence—Implications for a Critical Child Protection Studies

Lauren Elizabeth Wroe

Department of Sociology, Durham University, Durham DH1 3HN, UK; lauren.e.wroe@durham.ac.uk

**Abstract:** This paper presents data from a three-year, mixed methods study into the rate and impact of 'relocation' as a response to extra-familial harm in adolescence by children's social care teams. Participatory approaches to research design, data collection and analysis are used to gain insights from young people, parents/carers and professionals about the impact of relocations on safety. Professionals and young people report a range of harms implicated in the use of relocations, whilst sharing that the intervention often increases safety. Data are analysed zemiologically to understand this ambivalence, connecting micro accounts of harm with meso, institutional and macro structures that determine child protection intervention. Zemiology is put forward as a promising approach for a Critical Child Protection Studies.

**Keywords:** child protection; relocation; children in care; adolescence; extra-familial harm; zemiology; social harm





## 1. Introduction

This paper applies a 'zemiological' lens to data from the second phase of a three-year, mixed methods study exploring when, why and how frequently the UK child protection system uses 'out of area' placements ('relocations') to safeguard adolescents from harms they have experienced in contexts beyond their families. Data from phase one of the research have been previously published (Firmin et al. 2021). Here, data are analysed 'zemioloigcally' using the typology proposed by Canning and Tombs (2021). 'Zemiology' is proposed as a theoretical and analytical approach to advance a field of Critical Child Protection Studies that explores the impact of child protection interventions on young people's experiences of harm and safety *and* the contribution of child protection to producing harms.

Relocation of adolescents is a common child protection response (Firmin et al. 2021) to harms such as peer-on-peer abuse and sexual and criminal exploitation, known in the UK context as 'extra-familial harm' (EFH herein, HM Government 2018). Relocation can rely on the full legal force of child protection legislation to remove children from their families and place them in the care of the state at a significant distance from their communities (Firmin et al. 2021). In many cases, relocating young people involves statutorily, or at least practically, depriving them of their liberty (Roe 2022; Firmin et al. 2021). To date, analysis of when, how and why relocations are used to safeguard adolescents has concerned itself with dilemmas of professional decision making in multi-disciplinary professional contexts and the extent to which the intervention achieves safety for young people (Firmin et al. 2021). The recent 'Case for Change' interim findings from the Independent Review into Children's Social Care in England (MacAlister 2021) suggest that relocations indicate a failure to safeguard adolescents in extra-familial contexts, and participatory research with professionals involved in the relocation of adolescents indicates that relocations are often felt to be the only means of keeping young people safe whilst they are ambivalent about their use (Firmin et al. 2021).

Zemiology is an emerging discipline studying harm to people, communities, and the environment (Canning and Tombs 2021). Although the term zemiology precedes their work, Canning and Tombs (2021) make a case for 'zemiology' as a study of harm distinct from criminology and 'crime', and as a discipline *'that seeks to unearth harmful structures, policies, decisions and practices, evidences the impact that they have and thus generates radical and sustainable changes so that they can be mitigated or eradicated'* (Canning and Tombs 2021, p. 1). Whilst 'child protection studies' is conceivably the study of, or a discipline concerned with, protection, it is simultaneously and necessarily the study of harm, both those caused by abuse and those implicated in system responses. Empirical and anecdotal evidence suggests a limited or adverse impact of child protection interventions globally (see Bilson et al. 2017; Keddell et al. 2019); as such, there is a need for robust praxiological frameworks to explore not only the effectiveness of interventions in reducing abuse but the role of child protection systems in reproducing and/or creating new harms (Parton 2019).

Zemiology facilitates an analysis of macro-level harm through methods that produce micro-level insights and vice versa (Canning and Tombs 2021). This is the approach taken here. There has been limited application of zemiology to the field of child protection (discussed further below). The paper has three aims: to expand on and further contextualise previous accounts of the harms implicated in relocations of adolescents (Firmin 2019; MacAlister 2021; Firmin et al. 2021); to bridge everyday accounts of harm and the meso institutional and macro structural drivers that are often obscured from everyday experience; and to make a case for zemiology as a useful theoretical approach in the field of Critical Child Protection Studies.

## 2. Background

### 2.1. Relocation: A Statutory Child Protection Intervention into the Lives of Adolescents

Initial findings from the national Independent Review into Children's Social Care in England note the possible deleterious consequences of relocating adolescents, including missing episodes and child death (MacAlister 2021). At the same time, findings from phase one of this study (Firmin et al. 2021) indicate that approximately 1 in 10 under adolescents who are known to children's social care teams in England and Wales due to risk in extra-familial contexts are relocated. The findings report significant variability in the use of relocations among local authorities, who were grouped according to the rate at which they relocated young people: 'group one' relocated 0–5% of young people; 'group two' relocated 5–10% young people; and 'group three' relocated 10–25% of young people (Firmin et al. 2021). This variation was dictated by a strategic (or lack of) vision around how adolescents are safeguarded and what this means for distance placements. Group one services had invested significantly in keeping or bringing children home. Group two areas tended to have no strategy regarding adolescent distant placements, with professionals reporting that they were often used as a 'last resort' in cases of escalating physical risk. For group three, relocation was seen as a solution to adolescent risk, often in the absence of alternative placements, pathways or support offers for young people (Firmin et al. 2021). Follow-up interviews with professionals suggested a range of negative impacts of relocations, including disruption of relationships and impacts on young people's mental health and well-being (Firmin et al. 2021).

This paper analyses data from the second phase of this research, where young people, their parents/carers and professionals were asked about their experiences of relocation, what helped, and did not help, and the extent to which relocations created safety in their lives. This paper adopts a broader theorisation of harm beyond that which is defined as abuse to support an understanding of the impact of relocations, and of the ambivalence evident in the accounts of those who have proposed and have lived them.

### 2.2. Zemiology: A Theoretical Basis for Critical Child Protection Studies

The acknowledgement in the Case for Change (MacAlister 2021) of 'relocation' as a significant child protection intervention with a limited evidence-based begins (although in

a limited way) to situate 'relocation' in the macro-context of contemporary child protection debates, policies and practices. In the Case for Change, relocation is situated as a symptom of a confused child protection system that, with a traditional focus on families, is limited in its knowledge, resources and confidence when responding to adolescent harm. However, the issues identified in the report in relation to adolescent safeguarding mirror those identified in traditional family work: competing policy priorities (including support or criminalisation of young people/families), racial disproportionality, assessment without support and reliance on proceedings to remove children into the care of the local authority to manage risk (MacAlister 2021). This suggests, perhaps, that the issue is not that the child protection system is not designed to create safety for adolescents but that it is not designed to create safety at all, a view recently debated by others (Garrett 2021; Maylea 2021).

Zemiology as a discipline has emerged from Critical Criminology and Social Harm Theory. Critical criminology, and further Social Harm Theory, deconstruct 'crime' and 'criminality' as social and political constructs disproportionately used to control poor and racialised communities. It contends that many harms are not crimes and that many crimes do not cause significant harm. At the same time, many legal activities, policies and decisions (often carried out by powerful states and corporations) cause very substantial harms to people and the environment (Pemberton 2015). Therefore, crime is not a proxy for harm or vice versa. Ultimately, social harm theory contends that 'crime' reduction is ineffective, with criminal justice responses such as imprisonment and offender programmes doing little to rehabilitate those who have harmed (Hillyard et al. 2004). Zemiology is concerned not only with the ineffectiveness of crime reduction responses but with harms beyond crime, particularly those implicated in the activities of large-scale corporations and institutions, including, but not limited to, the criminal justice system. In this sense, zemiology bridges structural, meso-institutional and everyday experiences of harm (Canning and Tombs 2021).

Canning and Tombs (2021), whose call to 'do zemiology' is taken forward here, propose a provisional zemiological typology of harms:

- *Physical harms*: including murder, abuse and sexual misconduct, however, social harm theorists have stressed the sometimes distributed and indirect nature of these harms. For example, the link between preventable deaths through malnutrition, poisoning or environmental degradation and corporate and governmental practices or policy decisions (Pemberton, in Canning and Tombs 2021, p. 71);
- *Emotional and psychological harms*: including mental illness, but also issues such as sleeplessness and worry. Zemiologists consider the impact of indirect harms, such as collective trauma or anxiety resulting from local environmental degradation or public discourses on 'terrorism', for example (Canning and Tombs 2021). Importantly Canning and Tombs propose (drawing on the work of Herman 1992) that the causes of emotional and psychological harms are often identifiable only if those experiencing them can know and speak of those causes;
- *Financial and economic harms*: including personal or household financial loss (financial harm) or economic harms related to macro-economic mismanagement by corporations and governments and the impacts of poverty and inequality (Canning and Tombs 2021);
- *Cultural harms*: including harms to culture (i.e., destruction of culture), by culture (i.e., imposition of culture) and cultural harm as misrecognition (i.e., misrepresentation expanded upon in 'harms of recognition' below) (Boukli and Copson 2020, in Canning and Tombs 2021, p. 74);
- *Harms of recognition (or relational harms)*: imposition of an identity on a person/s that is 'spoiled' or 'blemished' and that functions to 'other' (Pemberton 2015, in Canning and Tombs 2021, p. 79) or 'subordinate' (Fraser 2000, in Canning and Tombs 2021, p. 79);
- *Autonomy harms*: including restrictions on access to social opportunities (including resources, education, employment, training and work) that limit social opportunity and ability to self-actualise (Pemberton 2015, in Canning and Tombs 2021, p. 79).

### 2.3. Zemiology and Child Protection/Harm

Some have already begun to apply a zemiological lens to the analysis of child protection systems and/or of harms in the lives of children, although the discipline is an emerging one. Parton (2019) proposes that Social Harm Theory can be usefully applied to child harm, recognising issues such as child maltreatment and neglect as tied to inequalities, both within families and at a societal level, with inequality linked to family pressures and greater health and welfare issues across populations. Understood in this way, harms to children are less a matter of personal or interpersonal deficit or dysfunction but the direct or indirect consequences of social inequality (Bywaters et al. 2016). Social harm or zemiological analyses have also been applied to policing and social care interventions into drug dealing impacting young people (Mason 2020) and 'county lines' (Wroe 2021). This research identifies a range of racialised social harms enacted by the state through covert and surveillance-oriented interventions. Featherstone et al. (2021) have applied a social harm lens to the analysis of child protection post the COVID-19 pandemic, aligning the approach with initiatives within the fields of child inequalities (Bywaters 2020) and child harm (Featherstone et al. 2018) to propose a post-pandemic social work that is more collaborative, restorative and hopeful. Zemiology has also been applied to children's experiences in the digital world as a means of proposing democratic alternatives to internet 'blocking' in schools (Hope 2013) or disciplinarian approaches to 'sexting' (Lee and Crofts 2013).

There is a significant body of evidence that speaks to the limitations of contemporary child protection systems, both in their ability to identify and prevent harm (Bilson et al. 2017) and their disproportional application (Bywaters et al. 2017) across racialised and working-class populations. Zemiology offers a theoretical framework to foreground the role of social work in *producing* harm and a conceptualisation of these harms as potentially *as* significant in the lives of young people as those caused by interpersonal 'crimes' and abuse. Participatory research carried out with children in South Wales who had experienced harm and had encountered the youth justice system (Haines and Charles 2019) showed that children rejected the label of victim, felt unsupported by services, and interestingly, adopted what the authors describe as a 'zemiological' account of harm whereby their understanding of what was a harm was markedly different from criminological and adult understandings. Whilst young people in the study were generally not well informed about the workings of the criminal justice system, their accounts evidenced tacit acknowledgement that these were services under strain and that this was impacting their experiences.

Following this work, this paper applies a zemiological lens to young person, parent/carer, and professional accounts of relocations to describe the full spectrum of harms implicated in their use and to bridge these everyday accounts with their macro-structural contexts. In doing so, it supports an understanding of the divergent and ambivalent attitudes towards the use of relocations (Firmin 2019; Firmin et al. 2021; MacAlister 2021).

## 3. Methodology

The data presented in this article were collected as part of a three-year mixed methods study exploring the rate, cost and impact of relocation as a response to adolescent extra-familial harm in England, Wales and Scotland between 2019 and 2021. The research was organised around two phases. Data from a quantitative survey (phase one) distributed to 15 local authorities to ascertain the rates at which they relocated adolescents exposed to extra-familial risk has been published elsewhere in a findings paper, as well as a thematic paper exploring the impact of relocations on relationships (Firmin et al. 2021; Firmin and Owens 2022). The methodological and analytical approach for phase two is described below.

### 3.1. Participants

Fifteen local authority children's social care teams participated in a phase one survey about the rate at which they relocate adolescents exposed to extra-familial risks (see Firmin et al. 2021). All 15 local authorities were approached to take part in the second phase of the study, which aimed to explore the impact of relocations on young people's experiences

of safety. Three local authorities were short-listed to take part in phase two; each had different rates of relocation (i.e., high, low or medium, as described earlier). A single point of contact (SPOC) in each service was asked to identify three young people (aged 10–25) who had experienced a relocation in the past year. SPOCs acted as 'gatekeepers' and used videos and consent forms provided by the research team to inform young people about the research and invite them to participate. Each young person was asked if they were happy for their parent/carer to be interviewed and to identify two professionals to be interviewed. Recruitment and data collection took place during the COVID-19 lockdown, and this limited recruitment in some areas. Due to under-recruitment from the local authorities, additional interviewees were recruited from a residential children's home (one parent, two young people, three professionals) and a national parent advocacy organisation (one parent, one professional). In total five young people (aged 13–18), three parent/carers and 15 professionals were interviewed (professionals included social workers (n = 11); residential children's home manager (n = 1); parent advocate (n = 1); youth engagement worker (n = 1) and head of Pupil Referral Unit (n = 1).

*3.2. Data Collection*

Interview schedules were designed in collaboration with young people, parent/carers and professionals. Young person interview schedules were designed in collaboration with a young researcher's advisory panel based at University of Bedfordshire, who have expertise in care placements and child harm. Two focus groups were held where feedback was sought about the questions and design of the interview. The feedback was integrated, and an activity-based interview pack was professionally designed. A pilot interview was carried out with one young person from the advisory group.

The parent/carer interview schedule was drafted by the research team, mirroring the structure of the young person interview, and then discussed with three parents from a national parent advocacy organisation via video or telephone call. Feedback was integrated into the final design of the interview schedule.

The professional interview schedule was drafted by the research team, again mirroring the design of the other two schedules, and was discussed in an online focus group with a Research Advisory Group (RAG) attended by key policymakers, professionals and academics.

Interviews were carried out remotely and asked participants to share their experiences of relocation, their views on the process and their perceived impact on safety. Additional grant funding was successfully applied for to fund support services that had an existing relationship with participants to provide 'wrap-around' preparatory and debrief support to young people and parents/carers who participated in the interviews.

*3.3. Data Analysis*

Interview data were thematically analysed (Braun and Clarke 2006) by the research team against a coding framework that corresponded to the research questions:

- The extent to which relocations were helpful/unhelpful;
- The perceived impact of relocations on safety (a holistic account of safety was adopted based on Shuker's (2013) multi-dimensional model of safety, including: physical, relational and emotional safety).

Initial themes were discussed in two focus groups, one with the young researchers advisory panel and one with the RAG. Feedback was integrated, and a final set of themes across the three datasets (young people, parents/carers and professionals) was identified under the headings 'effectiveness' and 'impact on safety'. A findings paper detailing the results of this thematic analysis is published elsewhere (Wroe et al. 2022), and a brief overview is provided below.

In this paper, the data are analysed using the zemiological typology of harms described above, linking participants' everyday accounts of safety and harm with meso, institutional and macro power structures.

*3.4. Ethics*

The research was granted ethical approval University of Bedfordshire. Consent was sought before, during and after all interviews in line with best practices (Whittington 2019).

*3.5. Limitations*

COVID-19 limitations resulted in fewer young people and parents being recruited than intended. This impacts the generalisability of the findings.

**4. Findings**

Initial analyses of the data surfaced a shared set of conditions for relocations that young people, parents/carers and professionals felt were important for a relocation to be effective. These were: quality and consistency of support, suitability of placement and planning. In addition, young people and parents identified two additional themes that were not raised by professionals: communication and decision making; and supporting relationships with family. These findings indicated the importance of asking about safety when planning a relocation and that professional priorities may overlook the needs of families. Initial analyses of the data indicated that when the conditions for effective relocations are not considered, relocations can achieve a reduction in physical risk whilst significantly undermining safety in other areas of the young person's life. These initial findings are discussed in full in a forthcoming article (Wroe et al. 2022).

Here, a zemiological analytic lens is applied to the dataset to further explore the range of harms reported by young people.

*4.1. A Zemiological Analysis of the Impact Relocation*

4.1.1. Physical Harms

As identified in phase one (Firmin et al. 2021), relocations were generally intended to address the escalating physical risk to young people that could not, or could no longer, be managed locally. Here, escalating and significant risks to the young person's physical safety were always the threshold and driver for the move. However, accounts from young people, their parents/carers and professionals revealed that relocations carried with them several risks of physical harm beyond instances of abuse. In a minority of cases, young people experienced direct physical harms because of the relocation, including experiencing physical restraint by professionals in their placement.

Several indirect physical harms were implicated in the relocation of young people, specifically resource constraints that undermined the effectiveness of social care intervention. These can be understood as physical harms indirectly linked to the decision to relocate and the wider policy and economic context in which these decisions are made. Professionals reported a lack of alternative options for keeping young people safe, noting that in many instances, the source of the harm was left unaddressed:

> *the thresholds for intervention are so high that usually it is years of abuse that's taken place before relocation is considered, and often there are no real alternatives to that.*

(Professional Interview)

Professionals were concerned about the low availability of placements (n = 6), the cost of placements (n = 2), lack of resources for specialist services as an alternative to relocations (n = 2), rushed decision making (n = 1), pressures on social care (n = 1), including from the police (n = 6), lack of other support services (n = 3) and placements having to end, or multiple placement changes, due to financial constraints (n = 1):

> *It's really patchy, to be honest, I would say. I would say some agencies are really great. A lot of parents that I've worked with, they would go and look at different units and Ofsted reports, but at the end of the day it depends on what's available, it depends on what the local authority's willing to pay. And often there's a step process that I think's quite challenging sometimes for parents, where they will go through, they'll try this, the cheapest one first.*

(Parent Advocate Interview)

Importantly, these constraints were seen to undermine the ability of the placement to achieve physical safety for young people (n = 8 professionals), with a lack of collaboration with young people, short timescales for decisions, unsuitability of placements that did not match need, lack of support in and around the placement (including support to maintain family relationships) all thought to increase the likelihood of the relocation failing and/or of young people going missing from their placement and being subject to further physical harms:

> *Did you ever leave it at some point?*
>
> *Aye!*
>
> *Why did you want to leave? Why did you want to do that?*
>
> *I dunno, I just didn't feel like it was working so I removed myself from the situation instead of talking about it.*
>
> *You were worried?*
>
> *Aye, like [inaudible 0:25:50] didn't want to be in the situation so I ran away from problems instead of facing it.*

(Young Person Interview)

These direct and indirect physical harms can be understood, zemiologically, as connected to neoliberal ideological and economic decisions that mean social care services deliver individual case-work models that target children's choices and behaviour (Lorenz 2016; Featherstone et al. 2018), leaving unaddressed the contexts in which they are harmed (Firmin 2019). As Firmin (2019) proposed, relocation locates risk *in* rather than *around* the young person, and the choice to do so carries with it many indirect consequences for the physical safety of young people.

### 4.1.2. Emotional and Psychological Harms

In some instances, young people's emotional and mental well-being significantly deteriorated in their out of area placements, and many young people reported feelings of isolation, displays of anxious behaviour and worry and uncertainty connected to their experience of being placed out of area:

> *[young person] was being secured because they were at risk of suicide or death through misadventure, that was the grounds for [young person] to be . . . But what we were acknowledging was their dysregulation and their high levels of distress was a consequence to their child sexual exploitation, a couple of weekends prior to that. Then the rejection from their family and then this young person came in, seen the relationships that she had, [young person] just spilled over, they couldn't contain all the different feelings and they completely spilled over.*

(Professional Interview)

> *there was a feeling that his mental health took a dip and there were feelings of isolation. He was picking at his eyebrows and his eyelashes, so there was nerves there. He was a teenage boy who was used to being out with his friends and participating in many different things who was then, with the best will in the world, with two staff who worked tirelessly with him. But [professional] and [professional] aren't as fun as what you'll, any of the 15-year-olds you're going to be at that point in time. So, it was about how we kept him occupied and focused and then there was an issue about accommodation.*

(Professional Interview)

Significantly, participants connected these experiences to process issues including how decisions were made and communicated. Young people linked opaque processes with experiences of discomfort, lack of control and distress:

> *I hate when, there's only one thing I ever hate, and it's when people promise me something then they don't do it, know what I mean.*

*Yeah. Yeah I do.*

*So it gets to me. I apologise. I meant to say that in the first part.*

**No that's fine. You don't need to apologise at all. And would you say that's the case for everyone, like professionals you've worked with, family, friends, all of them?**

*Aye. Aye.*

**Yeah.**

**You froze a bit on my screen just then. Can you say that again, what you just said, because I didn't hear it?**

*What did I say there? What did I just say? I've got other things going on in my -*

**You've got a load of things going on in your head. I said, so is that true for everyone; friends, family, professionals, promising you things and not delivering them?**

*Aye that, I hate when that happens. They promise you something and then they don't do it do it.*

**Yeah.**

*. . . got a good bond with. It gets to me, know what I mean like. It's not just a wee thing. It might seem like a small thing to a lot of people, but I don't think it is.*

(Young Person Interview)

*the people who are in charge of you know what's going on but they don't tell you until after they've put you in it, so they don't tell you until it's too late*

**Is that how you felt sometimes?**

*Aye, you feel like you put in to something, they don't explain what it's like and how it is . . . until you're already there.*

(Young Person Interview)

Zemiology supports an understanding of these distributed psychological effects. Young people displayed or reported a range of psychological impacts, including emotional distress and unease to mental illness, including re-traumatisation and depression. As Canning and Tombs (2021) note, the causes of emotional and psychological harms can be obscured, particularly because victims of harm are often silenced interpersonally, societally and politically. Zemiology, they argue, can create the conditions for those who have been harmed to understand the causes of harm and to name and voice them. Feminist movements have long taught the importance of connecting micro instances of harm with macro structures, and recovery from trauma can be facilitated by being able to identify and name both the interpersonal and the structural causes of violence (Herman 1992, in Canning and Tombs 2021). Whilst a causal relationship could not always be established (nor was it always reported by participants) between the relocation and emotional distress or mental ill health, two of five young people reported that they were not able to discuss their mental health concerns with any professionals involved in their placement, and for one young person multiple relocations were delaying their access to Child and Adolescent Mental Health Services (CAMHS):

*I mean I think that's difficult because I think for a lot of our young people that are running lines there is a reality that if you're local you've got more resources available, resources that you have control over, because the local authority has its resources, you would hope. So the boy in [new area] had to wait 12 weeks for CAMHS, even though he had the seven day follow-up, but even though he was immediately suicidal he waited months for an appointment with a CAMHS worker. He started seeing that person and then he moved back, and we've been told there's a nine-month waiting list for CAMHS in [hometown].*

(Professional Interview)

*I think for young people that do have significant ties to [local area] and in terms of relationships with family, then it's that sort of further loss of identity, loss of belonging that impacts on your self-confidence, your sense of worth, your general emotional health, and low mood and depression and anxiety.*

(Professional Interview)

Zemiological approaches to understanding emotional and psychological harms could create space for young people to talk about the emotional and mental health impact of child protection interventions and the wider policy and resource issues that determine the type and level of support they receive. For example, the lack of alternative support (as expressed by professionals) contributing to their relocation and under-funded mental health services.

### 4.1.3. Financial and Economic Harms

As with all social harms described in a zemiological typology, financial and economic harms often incorporate other forms of harm, '*including the mental or emotional effects of job or property loss, and the relational harms which families or social units living together can experience when financial pressures push them apart*' (Canning and Tombs 2021, p. 73). One parent interviewee described losing her secure tenancy as a result of relocating her family. In this instance, she had sought support from children's services to move her family due to significant physical threats to her son and ultimately decided to leave her family home. This parent described how she lost access to her furniture, how the Council struggled to find housing for her large family and how her family were eventually split up, with some family members living in a caravan. For this parent, the financial harms are evident; however, wider economic processes resulting in a lack of resources for services and availability of suitable accommodation (as expressed by professionals below) can also be understood zemiologically as contributing to this family's loss of tenancy and separation.

Zemiology provides a framework for describing the harms implicated in broader governmental, corporate and policy decisions that create inequality, poverty and austerity at a local and global scale. Current economic conditions, where funding for children's social care has seen a 24% reduction in the past decade (48% for early intervention services, Williams and Franklin 2021), are the context in which practitioners and families were navigating decisions made (or made for them) about a relocation. These economic harms are evidently linked then to other forms of harm experienced by young people:

*And the placement worked really well for her ( . . . )unfortunately the placement broke down, not due to any sort of reasons with the young person, it was more a financial issue really I think, because it was a solo placement for [young person], it was actually a two bedded unit, and we couldn't match what they were asking in terms of keeping that other bed open.*

(Professional Interview)

*We're paying for two properties, the finances had gone awry, we have no resources that can bridge those two boroughs. And that was because they were placed in safe accommodation and not supported to go through the other local authority's own homelessness route ( . . . ) If they'd been supported to do that, they would have got a service from [Local Authority] social care. But the way they were housed really limited the support available to them.*

(Professional Interview)

### 4.1.4. Harms of Recognition (or Relational Harms)

Social harm theorists have described 'relational harms' as '*enforced exclusion from social networks or personal relationships*' caused by 'social structures' that limit 'self-actualisation' (Pemberton, in Canning and Tombs 2021, p. 78). Often the purpose of relocation is to disrupt abusive relationships and to remove young people from harmful contexts (Firmin et al. 2021). At the same time, professionals, and more so young people and parents, have shared the ways in which these moves sever safe and protective relationships and bonds with family, friends, professionals and social networks (see also Firmin and Owens

2022). When asked about the impact of relocations on safety, participants, much like in work described by Haines and Charles (2019), responded with a zemiological account of harm that extended beyond the ability of the placement to manage risk or not, including specifically the impact on relationships:

> *I think it can make a young person more vulnerable, and breaking down those supports that they already have, maybe, you know, the dinner lady that served them dinner every day for the last five years, as a social worker, you know, you might not even be aware particularly of that relationship if it's not something the young person has discussed with you. But it could be that person that just smiles at them every day when they're giving them their dinner that is making a huge impact on that young person.*

> *So for me I feel it can be quite a dangerous practice moving young people out of areas where they know where they've got pre-existing relationships, and even friends, pets, you know, I've moved children who have been so distraught about not living next door to the neighbour's dog. They'd go out and stroke the dog or something.*

> *So I think it's decisions that we can take, but there's a lot of information that we're perhaps not always privy to around positive relationships in terms of what may be safeguarding a young person even just slightly that you're destroying with maybe that choice.*

(Professional Interview)

For Canning and Tombs (2021), 'harms of recognition' can precede these relational harms. These include institutional 'misrecognitions' (Fraser 2000, in Canning and Tombs 2021, p. 79) that distort individuals' or communities' realities and function to justify subordination and ill-treatment. Certainly, young people shared that the decision to relocate them felt like punishment:

> *That was alarming as I was a young girl [details of harm removed for anonymity], and I got punished. I'm the one who got put in secure never mind I've got an [injury from an accident preceding the relocation -this section removed for anonymity] . . .*

> **And at that point, when you say it was you that was punished, what do you mean by that?**

> *I was moved around. I weren't around my family. I needed my family. I had just come out of a traumatic experience. My friend died and I got an injury. I needed my family, and I was getting moved around and a lot of changes going on what shouldn't have been happening at the time, what obviously is getting spoken about now, and is getting chased up because certain things shouldn't have been going on with me being in the care system. [Inaudible 09:21] chase up every individual [inaudible 09:24] who they are.*

> **I'm sorry to hear about that experience. It sounds like a really hard experience, and then I hear what you're saying then; that then by being moved around you felt like that was a level of punishment then.**

> *You're alone when, you're alone when you're in another city. You have got these staff members, but really you're bringing up yourself in a city on your own. Like I've been in care homes and I've seen a lot. I've been in a secure unit where staff let certain kids spit in each other's drinks and pass it [inaudible 10:03]. That's the sort of stuff I've seen. Do you know what I mean? There's certain things always stick with you. It'll always stick with you. And that's what the system needs to understand, that us kids will always have that instilled in us.*

(Young Person Interview)

Young people's descriptions of being 'punished' can be read as testimonies of their misrecognition by professional agencies. The young person above shares that she was a 13-year-old girl, foregrounding her youth and details of her victimisation, and contrasts this with the response she received—punished and placed in a secure (welfare) provision (a 'secure' placement requires a court order that prevents a child from leaving). In this

sense, the young person could be understood as having been 'misrecognised' as somehow culpable for her abuse and treated accordingly.

Whether professional agencies intentionally responsibilise adolescents for the harm they experience in extra-familial contexts, a zemiological analysis can support an understanding of how pervasive attitudes towards teenagers as troublesome or as comparable to adults in their choices works to exclude adolescents (particularly Black and working-class adolescents) from narrowly defined recognitions of victimhood (Brown 2019; Davis and Marsh 2020). The stigmatisation of adolescents in this way could work to implicitly rationalise their treatment as 'criminals' (Tyler 2021), with them being removed from families and friends in a system ostensibly geared towards their safety:

> *So, you've recently moved in to secure, right? So, that's another move then is that your third move?*
>
> *Yeah.*
>
> *And, how was that like compared to the others?*
>
> *It was not good.*
>
> *It's not good? Why?*
>
> *No, it's like you're locked up, it's like a jail.*
>
> *Like a jail … so, like for the other moves, did you understand why you had to move?*
>
> *Aye.*
>
> *But do you agree with it?*
>
> *No.*
>
> (Young Person Interview)
>
> *"You're doing this to me, moving me away from all my friends, you've put me in prison, you've cut me off." I was like the next step is secure, and I don't want to put him in secure, because secure doesn't work either.*
>
> (Professional Interview)

### 4.1.5. Cultural Harms

Several young people reported feeling 'out of place' in the area to which they had been relocated. For some, the familiarity or unfamiliarity of the local high street (i.e., what sorts of shops and supermarkets were there) was important. For others, the décor in the accommodation led to feelings of unease and unfamiliarity, which led to them asking to leave, and one young person reported that they felt they had to change their persona to fit in in different settings:

> *I'm from up north. I went to [city] and how they are up there and how it is here is two different things You have to change your whole persona. Then you get moved to another city down south, so you have to change your persona again. Then you get moved here so then you have to change again. It's literally like …*
>
> *And so can you explain that a little bit more to me. So in what ways do you have to change your persona, and why do you think it is that you need to do that?*
>
> *Because different cities and different, like south, north, east, west, different places have different attitudes and different approaches, and different talks, different like code talk. Not even code talks, but different banter. Everything's different. So I'm having to adjust and keep up with where I'm being moved, and that's with anyone who's been moved out of area, change and keep up to the point where you end up being this angry person. You end up being this angry person because you're having to try and keep up with any rules and having to adjust [inaudible 11:49] it happens again, and you're always ready for it.*
>
> *Ready for the next move?*

*Yeah, you're always ready. When you're in care you're sat and you know that you'll move again. You can never be too comfortable in care. You can never, this is going to be my home for the next three years. It's never like that.*

(Young Person Interview)

These expressions of unease and unfamiliarity could be understood as 'cultural harms' where young people had to adapt their accustomed ways of doing and *being* to get by, removed from the familiarity of their own family and community cultures.

Young people from working-class and racially minoritised families are disproportionately likely to have social work involvement in their lives and to be placed away from their families (Bywaters et al. 2017). Previous findings identified 215 young people who were placed out of area due to extra-familial risks in September 2019 (Firmin et al. 2021). A total of 19% were Black, and a further 14% were identified as 'Mixed', Asian (3%) or 'Other' (5%). As only 5% of under 18's in Britain are Black, an over-representation of Black children in out of area placements is indicated. This would reflect and exceed the overall children in care statistics where Black children are over-represented as children in care at a rate of 7% (HM Government 2021). A zemiological analysis, therefore, flags that further monitoring of practice data is required to understand any disproportional use of out of area placements for Black children and could make a case for these indicative figures to be appropriately considered in Britain's post-colonial legal/institutional context. Such figures may reflect the persisting disproportionality in national trends for children in care and in prison settings, where Black and 'Mixed' ethnicity children are significantly over-represented in the youth justice estate (51%) and amongst children referred to secure children's homes on welfare grounds where they experience differential treatment (Roe 2022).

### 4.1.6. Autonomy Harms

Relocations warrant monitoring and investigation of their use as they involve significant acts of state intervention into the lives of young people and their families. A total of 90 out of the 215 young people identified as relocated in phase one (Firmin et al. 2021) were relocated under a Care Order (Section 31), a statutory order that places the child in the care of the local authority with parental responsibility shared between the local authority and the parents. At least two professionals interviewed in phase two recognised that the relocation equated to a 'deprivation of liberty' (these were secure welfare settings) and spoke of the detrimental impacts on mental health. Critically, where deprivations of liberty are considered for older teenagers, they must be the least restrictive option (Roe 2022). Where alternative means of supporting adolescents at risk in extra-familial contexts are not resourced (as suggested by professionals in this study), this poses ethical and legal questions about the grounds on which deprivations of liberty are ordered and the extent to which alternatives are or can be explored. Research undertaken by (Roe 2022) indicates a rise in applications to the courts to deprive children of their liberty in alternative placements (with demand for secure placements exceeding availability) and warns of a lack of oversight as to what sorts of restrictions are being placed on these children who often have complex needs and presenting risks, including criminal and sexual exploitation.

A zemiological analysis could situate the use of deprivations of liberty through relocations (whether formally recognised or not, being placed 100 miles from your hometown with no mobile phone is significantly restrictive) as a function of 'carceral' social work, and a long tradition of removal, separation and confinement being used by social workers in their work with families and communities. Jacobs et al. (2020) define carceral social work as that which collaborates with the police and policing and that which uses methods of social control outside of work with the police and criminal courts, in particular targeting poor and racially minoritised families. Certainly, young people interviewed here felt and spoke to the parallels between their experiences of relocation and being in 'jail' or 'prison' (as described above).

However, autonomy harms are not only those that directly restrict the movement and freedoms of young people; they are defined (Pemberton 2015, in Canning and Tombs 2021)

as harms to an individual's or communities' ability to reach their potential. Young people interviewed here spoke about the ways in which being relocated and then experiencing multiple moves between placements caused significant and long-term disruptions to their lives, reflecting wider trends in outcomes for care experienced children:

> *Because it feels like a game because you're in one place and then you start to have bonds with people and then you move and then you go, "I'm not trusting anybody" and then you start to open up and then you get moved again.*

(Young Person Interview)

> *You're alone when, you're alone when you're in another city. You have got these staff members, but really you're bringing up yourself in a city on your own.*

(Young Person Interview)

This was particularly apparent in relation to education and training opportunities:

> *For one I was, the thing where people need to realise and all the people need to realise, it starts from school. It all starts from being expelled or excluded from school. Do you know what I mean? When I got excluded I were in year seven. I was going out with older people because I wasn't at school and everyone else was. So I never had no one my age to go to school with, or chill with every day, because everyone else was at school. So that's where it all started; school. And the thing is now, because I got kicked out of school and I was tarnished in school from year seven, I've never had a good, I've never had [inaudible 25:01] education. It's always been a fight because I've got [inaudible 25:06] I act like this, or my paperwork says I can't do this. I actually can do this*

> **Right.**

> *and it's always been a battle -*

> **Right.**

> *- and now I'm 18 I'm willing to do whatever to be able to get my education, because that's the one thing what I missed out on throughout all my care. I was literally [inaudible 25:28] care, and I'm learning myself because there's no teachers available and stuff.*

> *I'm sat doing a B-Tech on my own, and do you know what I mean? [Inaudible 25:40] like that. I'm having to apply for education*

> **And do that for yourself.**

> *terrible, very terrible battle, but I really, really think that I could, that the system could have helped me a lot more [inaudible 25:55] I start college last year. My EHC plan was out of date by about three years [inaudible 26:04]. So they should be updating my stuff so I can do what I need to do, because I'm [inaudible 26:10] because my EHC plan isn't updated and that's my fault, and probably that's how I [inaudible 26:22] battling with the system and education [inaudible 26:26].*

> **It sounds like having to take it into your own hands sometimes.**

> *Yeah. I've got so much books of education books and stuff like that but I teach myself because I like learning, but the system won't let me into college or I'm too high risk to go to [inaudible 26:44], and that's what has happened, because of all the moves and this and that and whatever.*

(Young Person Interview)

> **Alright, and then when you moved here, was there anything that you wish was different about where you are? So apart from, like you said, all the -**

> *[Inaudible 21:58] didn't happen.*

> **You wish it didn't happen, and why not?**

> *Because I'd still be in college if none of this happened.*

> **Are you not in college at the moment?**

> *No.*

*But you want to be?*

*Yeah. I want to finish [inaudible 22:21] degree in construction.*

*Right. Do you know if maybe you'll have a chance to go back?*

*No*

*You don't think you will?*

*No.*

*So you wish you were back school*

*Yeah.*

*And do you have any kind of online school at the moment or anything?*

*No. I go to something at the youth centre.*

*And what's that like?*

*It's alright, but that's just to do with like other purposes and that.*

*Okay so it's not really about a degree.*

*No*

(Young Person Interview)

However, it is important to note that for several young people, the relocation created opportunities for education and for social engagement, meeting new (safe) friends and getting access to better local provisions (in one case, a better local youth club). They also created opportunities for young people to access therapy and to form supportive relationships with professionals away from crisis situations in their home communities; these tensions are discussed further below. Applying a zemiological lens to the views reported by young people, parents/carers and professionals could provide a framework for understanding the ambivalence and contradictions in these micro-level accounts of safety and harm in the context of institutional and macro-level economic, policy and practice structures.

*4.2. Ambivalence in Participant Accounts*

As a social worker and social work academic, it is often difficult to be confronted with the reality that, too often, attempts to help can involve significant amounts of hurting. At the same time, the accounts provided through this research evidenced that professionals (as well as, of course, young people and parents) felt unease, anxiety, uncertainty and in some cases, a sense of helplessness about the options available to them, and therefore young people and families, in their current practice contexts:

*And to me it feels really sad because we have upheld our service to say we are there to protect and serve these families, but it feels as if we've left them hanging out to dry.*

(Professional Interview)

*I definitely think sometimes when people come to me as a senior manager, asking for a young person to be placed away is absolutely we don't know what to do, and we're finding it very difficult to see this young person being draw into risk in front of our eyes. But there's also something which is about we don't know what to do. I get that, you know, I get it, I've been that person. So another risk assessment, another strategy meeting, another briefing, another . . . So I get it, there is something about we can't continue to let both in terms of the work it creates, and in terms of the impact on the young person.*

(Professional Interview)

*But I do feel in terms of specialist emotional health services for those young people that require them we are still a ways from that. It is around requesting those resources, the argument of funding for those specialist services, which for me, I feel they should be more readily available.*

(Professional Interview)

The data clearly indicates that professionals feel their options were limited due to the level of risk facing young people, but also due to impoverished service settings that are not resourced to help and where there remains a mismatch between how problems are framed and the solutions on offer. Despite these reservations, when asked about the overall impact that relocations had on safety, 10 out of 23 participants felt the relocation made them/the young person safer (9 professionals also shared that the relocation had not reduced risk or addressed vulnerability). This seemed incongruent, but in circumstances that were so constrained, it was evident from the data that relocation, with the many potential harms it could cause to young people's relationships and well-being, was the only option they felt they had to keep the young person physically safe:

> *So, it was needed. I don't agree with secure accommodation, but it was needed at that time and I think it's something that I'm going to take up with my senior management that we need to try and replicate a kind of secure environment without children having to go to secure, because it doesn't work for them. But we do need to have something whereby our young people can go to, to keep themselves safe, but it's still in their community with the trusted people around them. But that's above me.*

(Professional Interview)

Whilst professionals shared that restrictions and current practice models prevented them from providing the type of support they felt was needed, young people were less able to articulate why the intervention that they did not want and that felt hard in many significant ways was (they felt) the best thing for them:

> *It's been difficult sometimes.*
>
> **It has been?**
>
> *Aye. It's been a struggle man. First time I come in here I was only 12. It's been a struggle, but it's worked.*
>
> **So you were only 12 when you first came in, so that's a long time to get to know them over time and get to know who they are. And do you think it is just time that's helped, or has anything else helped?**
>
> *[Inaudible 19:26] know what I mean, because I've clicked with certain staff straightaway, and then I've took time to work with some staff to . . .*
>
> **And what was making it a struggle at times? What was that?**
>
> *Because when I first came in here I was just a small boy. I was running about daft and that, just basically causing it, and then obviously—and that, and I used to get off my head and then I'd get moved about . . .*

(Young Person Interview)

> **Do you think that at some point you wished that something was done differently?**
>
> *Yeah.*
>
> **You do or you don't?**
>
> *Like I know why I was put in, like all the different movements but I didn't want, not that I didn't want to, I knew that it was the best thing to do but I didn't want to meet new people like staff and stuff it was kind of . . .*
>
> **Okay, so you knew it was the best thing for you but you didn't want to have to keep meeting new people**
>
> *Yeah.*
>
> **Because is it like having to build all those relationships and trust again, do you find that really difficult?**
>
> *Sometimes.*
>
> **And, do you think there are things that would have made the experience of moving a bit easier for you? Like what would have helped make it better?**

*Knowing what to expect*

(Young Person Interview)

The accounts shared here indicate a definite lack of choice and alternatives. Whilst professionals spoke to the resource deficient contexts they were working in, young people, whilst reporting a spectrum of social harms connected to child protection help (beyond the abuse they were experiencing), were seemingly less privy to, or able to articulate, this wider context. If young people are excluded from decision making (as they expressed above) and from understanding the macro, structural issues that drive institutional and professional decision making, their ability to understand, contextualise and recover from their experiences of harm (broadly defined) are potentially undermined.

## 5. Conclusions

This paper does not intend to argue for or against the use of relocations to safeguard adolescents from risks in extra-familial contexts. It is evident from the small dataset analysed here that relocations can have a broad range of supportive and harmful impacts on young people, frequently leading to increased feelings of safety whilst resulting in significant disruptions to important (safe) relationships and education, accompanied by varying (and at times very significant) levels of emotional distress and mental ill-health. Placing a child in care at a distance from their family is a significant interference by the state in private and family life, in some cases depriving parents of full parental responsibility and young people of their liberty. In all cases, guidance states that child protection interventions, such as relocations, should only be used where they are in the best interests of children and other options of safety have been exhausted (Roe 2022; Firmin et al. 2021). This threshold is undermined in a context where there are felt to be limited or no other options.

The data presented here are from a research study that aimed to use participatory approaches to research design and data collection to understand what we learn about relocations when we engage young people, their parents/carers and professionals in conversations about the impacts of relocations on safety. Analyses of young person, parents/carer and professional interviews surfaced a range of harms beyond those that would typically be considered (criminal) harm/abuse or professional malpractice. This paper aimed to think zemiologically about these harms.

The young people who shared their experiences of relocations had all experienced or been at risk of serious physical and/or sexual harm. In many instances, they were relocated when social workers no longer knew how or had the resources to protect them from these harms. For professionals, their constrained practice contexts (as well as high levels of physical risk to young people) contributed to 'last resort decisions' to relocate (see also Firmin et al. 2021). The analysis presented above demonstrated how these last resort decisions had significant impacts on the lives of young people who, much like in the research by Haines and Charles (2019), to some extent, understood that they had experienced a broad 'zemiological' range of harms that exceeded their experiences of (criminal) abuse, and professional malpractice.

Canning and Tombs (2021) work lays out a proposal for 'doing zemiology' that includes documenting harms alongside an analysis of how they are produced. In this paper, professionals spoke to the institutional, meso-level contexts that constrained their choices and abilities to provide safety for young people (in some cases, they acknowledged, causing harms). The analysis presented here has gone some way to linking these institutional constraints to macro, structural processes such as inequality, austerity and individualism that dictate how adolescence, violence and helping are understood in contemporary neoliberal society.

Canning and Tombs (2021) propose that zemiological enquiries centre on the use of language and dialogue to name and 'consciousness' raise about societal causes of personal and interpersonal harms, so that we can begin to have conversations about, and mobilise around, structural causes. The participatory approaches to research design, data collection and data analysis used here created opportunities for discussions about safety

and harms implicated in child protection interventions both in the design of the methods for data collection, through participant interviews, and by collaborative analysis of the data. Policy recommendations have been made based on these findings, as well as the publication of a set of resources for young people, parents/carers and professionals who are navigating proposed relocations. Framed zemiologically, future research could extend these approaches to create opportunities for young people, their families, and professionals to engage in conversations about macro power structures that inform their experiences of harm and help, alongside or beyond policy reform recommendations, to begin to imagine radical alternatives. This 'studying up' (Canning and Tombs 2021, p. 123) should be a central concern for Critical Child Protection Studies.

**Funding:** This research was funded by Samworth Foundation.

**Institutional Review Board Statement:** The study was conducted in accordance with the ethical framework approved by the Institute of Applied Social Studies Research Institute Ethics Panel of University of Bedfordshire IASR_04/19 7 February 2020 for studies involving humans.

**Informed Consent Statement:** Informed consent was obtained from all subjects involved in the study.

**Data Availability Statement:** Not applicable.

**Conflicts of Interest:** The author declares no conflict of interest.

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
