# Peer review of "When Helping Hurts: A Zemiological Analysis of a Child Protection Intervention in Adolescence—Implications for a Critical Child Protection Studies"

_socsci, doi:10.3390/socsci11060263_

Round 1

Reviewer 1 Report

Comments for Author(s)

Thank you for the invitation to review this article. I see merit in the topic and the approach using zemiology to consider harms experienced by adolescents who are relocated. 

Primary feedback is regarding the structure of the article

- there are a number of long run-on sentences that require adjustment

- frequent use of the word whilst  - the author should look for some other options

Section 2 - Background requires further separation with additional headings for clarity such as typology, relationship to child protection for example.

There are terms that need clarifying - wrap-a-round support, secure/locked settings - don't assume that the reader will know what these areas they are very specific to the child protection field.

Page 6 - you introduce the 8 cases (Case for Change) - this is confusing as it appears in the findings and reads as if it is related to this current study (it is not). Overall, there are several places where data/findings from Phase One of the study are included - report the findings from Phase 2 and then in summary you can speak to areas aligned between Phase 1 and Phase 2 - this will be easier for the reader to follow. 

Use of typology headings and format to report findings

- Keep the same reporting structure under each heading. In the Emotional and Psychological section, you appear to start with a quote - italicized. In other sections, you start with information and follow with the quotes.

Page 14 - New heading: Using zemiology to understand ambivalence in everyday accounts of child protection interventions. This section tries to summarize critical themes - sub-headings would provide clarity such as limited options, helplessness, safety versus risk (some examples to consider). The last part of this section with quotes from the youth might better be placed up in the typology as it feels like an add-on here. Some further considerations are:

- Paragraph 1 - some repetition from the introduction

- Portions of this section fit better in the conclusion - for example, paragraph 2 should be reviewed. 

Page 15 "This paper does not intend ... - some of what is in this paragraph would fit better in a Limitations Section.

Conclusion - in this section you again lead with the findings from Phase 1 - see earlier comments. These conclusions should be focused on the Phase 2 findings primarily.

This is a critically important topic as uncovering the intersecting nature of the issues is needed for the system to change in positive ways.

Author Response

Thank you for your comments, they have been really helpful. Please see tracked changes on the manuscript and responses below.

Reviewer One

Thank you for the invitation to review this article. I see merit in the topic and the approach using zemiology to consider harms experienced by adolescents who are relocated. Thank you.

Primary feedback is regarding the structure of the article

- there are a number of long run-on sentences that require adjustment I have proofread and edited the article again to remove long run-on sentences (see tracked changes)

- frequent use of the word whilst - the author should look for some other options I have proofread and edited the article again to remove repetition of ‘whilst’ (see tracked changes)

Section 2 - Background requires further separation with additional headings for clarity such as typology, relationship to child protection for example. Please see additional sub-heading ‘Zemiology and child protection/harm’ inserted.

There are terms that need clarifying - wrap-a-round support, secure/locked settings - don't assume that the reader will know what these areas they are very specific to the child protection field. I have read through the article and add clarification where I have used child protection specific terms (i.e., ‘secure’ and ‘wrap-around’)

Page 6 - you introduce the 8 cases (Case for Change) - this is confusing as it appears in the findings and reads as if it is related to this current study (it is not). Overall, there are several places where data/findings from Phase One of the study are included - report the findings from Phase 2 and then in summary you can speak to areas aligned between Phase 1 and Phase 2 - this will be easier for the reader to follow. Thank you for this feedback, really helpful. I have deleted reference to the phase one findings and the Case for Change.

Use of typology headings and format to report findings

- Keep the same reporting structure under each heading. In the Emotional and Psychological section, you appear to start with a quote - italicized. In other sections, you start with information and follow with the quotes. This appears to be a formatting error when the journal has moved the manuscript into their template. Please see start of Emotional and Psychological section, this is not a quote (should not have been italicised).

Page 14 - New heading: Using zemiology to understand ambivalence in everyday accounts of child protection interventions. This section tries to summarize critical themes - sub-headings would provide clarity such as limited options, helplessness, safety versus risk (some examples to consider). This section is not intended to be a summary but to demonstrate the ambivalence that participants shared toward the use of relocations. I have changed the sub-heading to indicate that this is still part of the findings section.

The last part of this section with quotes from the youth might better be placed up in the typology as it feels like an add-on here. See above, this section is intended to describe the ambivalence reported by participants and to suggest that professionals could contextualise with knowledge of the constrained practice contexts they are working in, something that wasn’t (necessarily) afforded to young people. I have made changes to this section to try to make this clearer.

Some further considerations are:

- Paragraph 1 - some repetition from the introduction – repetition noted, first sentence removed.

- Portions of this section fit better in the conclusion - for example, paragraph 2 should be reviewed. Please see comment above, this has been included here to frame a discussion about the contradictions and complexities in doing social work (including the authors’ own experiences)

Page 15 "This paper does not intend ... - some of what is in this paragraph would fit better in a Limitations Section. I have moved some of this to the conclusion. See tracked changes.

Conclusion - in this section you again lead with the findings from Phase 1 - see earlier comments. These conclusions should be focused on the Phase 2 findings primarily. I have removed reference to Phase One.

This is a critically important topic as uncovering the intersecting nature of the issues is needed for the system to change in positive ways. Thank you for this feedback and for your review.

Reviewer 2 Report

Why the different fonts for the participants comments? Maybe stick to the same for the young people, same for workers etc.

Knowing the age of the Young People in the study would be helpful. Perhaps more info about the participants would help. 

Would benefit from longer amounts of transcripts, especially from the young people - given this is all about them,

Page.10. The Haines and Charles (2019) quote then immediately followed by another quote is confusing. 

Page 13. What does YPNLZ and YRAP consultation mean? where did they suddenly come from?

Page 16. Conclusion. Para 4. Third Line - Capital S needed for 'so'.

Page .11. "I was a 13 year old girl" Young Person Interview. How old were they when interviewed for this study?

Page. 12. "Research (Roe) indicates should be "Research undertaken by Roe (2022).

Apart from this. Bloody Brilliant.

Author Response

Thank you for your comments, they were really helpful. Please see tracked changes on the document and responses below. 

Why the different fonts for the participants comments? Maybe stick to the same for the young people, same for workers etc. – Thank you for this feedback, the fonts have been changed so that they are consistent.

Knowing the age of the Young People in the study would be helpful. Perhaps more info about the participants would help. The ages of the young people have been added to the participant’s section. Details of the professionals have also been added.

Would benefit from longer amounts of transcripts, especially from the young people - given this is all about them – thank you, I have extended extracts from young people where it felt appropriate to do so, i.e., where the conversation had continued on the same theme and where including further information wouldn’t compromise their anonymity.

Page.10. The Haines and Charles (2019) quote then immediately followed by another quote is confusing. I am not sure what this is referring to but I think something funny happened to the formatting when the journal put the manuscript into their template. Please see section and hopefully the re-format has clarified this.

Page 13. What does YPNLZ and YRAP consultation mean? where did they suddenly come from? Apologies – this has been changed.

Page 16. Conclusion. Para 4. Third Line - Capital S needed for 'so'. edited

Page .11. "I was a 13 year old girl" Young Person Interview. How old were they when interviewed for this study? Young people’s ages added to participant section. I have actually now removed reference to this young person’s age from the quote as I want to ensure their anonymity given the small sample size and the extra detail I have now included by extending the extract.

Page. 12. "Research (Roe) indicates should be "Research undertaken by Roe (2022). Added, thank you.

Apart from this. Bloody Brilliant. Thank you!!